# Wastewater Surveillance of SARS-CoV-2 in Zambia: An Early Warning Tool

**DOI:** 10.3390/ijms25168839

**Published:** 2024-08-14

**Authors:** Doreen Mainza Shempela, Walter Muleya, Steward Mudenda, Victor Daka, Jay Sikalima, Mapeesho Kamayani, Dickson Sandala, Chilufya Chipango, Kapina Muzala, Kunda Musonda, Joseph Yamweka Chizimu, Chilufya Mulenga, Otridah Kapona, Geoffrey Kwenda, Maisa Kasanga, Michael Njuguna, Fatim Cham, Bertha Simwaka, Linden Morrison, John Bwalya Muma, Ngonda Saasa, Karen Sichinga, Edgar Simulundu, Roma Chilengi

**Affiliations:** 1Churches Health Association of Zambia, Lusaka 10101, Zambia; jay.sikalima@chaz.org.zm (J.S.); mapeesho.kamayani@chaz.org.zm (M.K.); dickson.sandala@chaz.org.zm (D.S.); chilufya.chipango@chaz.org.zm (C.C.); karen.sichinga@chaz.org.zm (K.S.); 2Department of Biomedical Sciences, School of Veterinary Medicine, University of Zambia, Lusaka 10101, Zambia; walter.muleya@unza.zm; 3Department of Pharmacy, School of Health Sciences, University of Zambia, Lusaka 10101, Zambia; steward.mudenda@unza.zm; 4Public Health Department, Michael Chilufya Sata School of Medicine, Copperbelt University, Ndola 21692, Zambia; dakavictorm@gmail.com; 5Zambia National Public Health Institute, Ministry of Health, Lusaka 10101, Zambia; mkapina100@gmail.com (K.M.); kunda.musonda@znphi.com (K.M.); joseph.chizimu@znphi.com (J.Y.C.); chilufyam2007@gmail.com (C.M.); otridah.kapona@znphi.co.zm (O.K.); roma.chilengi@znphi.com (R.C.); 6Department of Biomedical Sciences, School of Health Sciences, University of Zambia, Lusaka 10101, Zambia; kwenda.geoffrey@unza.zm; 7Department of Epidemiology and Biostatistics, School of Public Health, Zhengzhou University, Zhengzhou 450001, China; kasangaanita@gmail.com; 8Global Fund to Fight AIDS, Tuberculosis and Malaria (GFATM), 1201 Geneva, Switzerland; michael.njuguna@theglobalfund.org (M.N.); fatim.jallow@theglobalfund.org (F.C.); bertha.simwaka@theglobalfund.org (B.S.); linden.morrison@theglobalfund.org (L.M.); 9Department of Disease Control, School of Veterinary Medicine, University of Zambia, Lusaka 10101, Zambia; jmuma@unza.zm (J.B.M.); nsaasa@unza.zm (N.S.); 10Macha Research Trust, Choma 20100, Zambia; edgar.simulundu@macharesearch.org

**Keywords:** wastewater, surveillance, early warning, SARS-CoV-2, COVID-19, Zambia

## Abstract

Wastewater-based surveillance has emerged as an important method for monitoring the Severe Acute Respiratory Syndrome Coronavirus 2 (SARS-CoV-2). This study investigated the presence of SARS-CoV-2 in wastewater in Zambia. We conducted a longitudinal study in the Copperbelt and Eastern provinces of Zambia from October 2023 to December 2023 during which 155 wastewater samples were collected. The samples were subjected to three different concentration methods, namely bag-mediated filtration, skimmed milk flocculation, and polythene glycol-based concentration assays. Molecular detection of SARS-CoV-2 nucleic acid was conducted using real-time Polymerase Chain Reaction (PCR). Whole genome sequencing was conducted using Illumina COVIDSEQ assay. Of the 155 wastewater samples, 62 (40%) tested positive for SARS-CoV-2. Of these, 13 sequences of sufficient length to determine SARS-CoV-2 lineages were obtained and 2 sequences were phylogenetically analyzed. Various Omicron subvariants were detected in wastewater including BA.5, XBB.1.45, BA.2.86, and JN.1. Some of these subvariants have been detected in clinical cases in Zambia. Interestingly, phylogenetic analysis positioned a sequence from the Copperbelt Province in the B.1.1.529 clade, suggesting that earlier Omicron variants detected in late 2021 could still be circulating and may not have been wholly replaced by newer subvariants. This study stresses the need for integrating wastewater surveillance of SARS-CoV-2 into mainstream strategies for monitoring SARS-CoV-2 circulation in Zambia.

## 1. Introduction

The severe acute respiratory syndrome coronavirus-2 (SARS-CoV-2) was identified as the cause of the coronavirus disease 2019 (COVID-19) in the year 2019 [1,2]. Since the beginning of the COVID-19 pandemic, numerous mutations of SARS-CoV-2 have been identified. Periodic viral genomic sequencing helps to detect new genetic variants circulating in communities. A variant is recognized as a variant of concern (VOC) or variant of interest (VOI) by the World Health Organization (WHO) [3]. A VOC is a variant that presents many malignant changes and demonstrates an enhanced risk of transmissibility, virulence, or changes in clinical disease presentation, impairing the efficacy of diagnostic, preventive, and therapeutic options [4]. Evidence has demonstrated that different variants of SARS-CoV-2 have been circulating globally [5,6]. These variants have rapidly undergone mutations, making treatment and vaccination goals difficult to attain and potentially threatening further waves of COVID-19 [7,8,9,10,11]. Therefore, there is a need for the continuous development of newer and improved diagnostic methods, novel vaccines, and drugs to address COVID-19 [12].

Since its emergence and spread, many strategies have been employed to improve the surveillance of COVID-19 [13,14,15,16,17]. Wastewater surveillance has emerged as a way to monitor the presence of SARS-CoV-2 in communities as the virus is shed in wastewater [18,19,20,21]. This method can help identify SARS-CoV-2 shed in sewage and other wastewater types and could help to understand communities with a high prevalence of symptomatic or asymptomatic COVID-19 [22,23]. Additionally, detecting SARS-CoV-2 ribonucleic acid (RNA) particles in wastewater may indicate emerging COVID-19 clusters in a particular community [24,25]. Wastewater surveillance for SARS-CoV-2 involves monitoring wastewater for the presence of the virus to track its spread in a community using polymerase chain reaction and genomic sequencing detection methods [24,26,27,28]. This approach can provide insights into circulating variants of SARS-CoV-2, act as an early warning system for COVID-19 outbreaks, and help identify areas where the virus is spreading [26,29,30]. Wastewater-based surveillance should be employed as a complement to existing disease surveillance systems such as clinical surveillance [31,32,33,34]. In clinical surveillance, there is a reliance on symptomatic patients, which precludes asymptomatic individuals, thereby underestimating the magnitude of the disease [35].

During the early stage of the pandemic, a study in the Netherlands was able to isolate SARS-CoV-2 in wastewater close to a week before the first case of COVID-19, providing a proof of concept for the isolation of SARS-CoV-2 in wastewater and as an early indication of clinical onset in a population [36]. Furthermore, studies in Australia, the Czech Republic, and Italy demonstrated the presence of SARS-CoV-2 in wastewater obtained from water treatment plants, providing further credence to using wastewater samples for COVID-19 surveillance [37,38,39]. Another study in the USA found that wastewater surveillance was effective in detecting SARS-CoV-2 and contributed to the diagnosis of more than 85% of COVID-19 cases [40]. In Pakistan, a study found that 81.5% of SARS-CoV-2 was detected in wastewater [24]. This evidence shows that the presence of SARS-CoV-2 RNA in wastewater samples can be used as an early warning system for COVID-19 in communities [41,42].

In Africa, there are limited SARS-CoV-2 wastewater surveillance activities reported to be important in addressing COVID-19 [43]. A study conducted in South Africa detected SARS-CoV-2 in wastewater including the Beta, Delta, and Omicron variants [35]. In Ghana, the presence of SARS-CoV-2 RNA in wastewater was detected in 17% of the samples tested for the presence of viral RNA [44]. These studies confirmed that SARS-CoV-2 wastewater surveillance could contribute to an improved understanding of SARS-CoV-2 epidemiology and transmission dynamics and contribute to early detection, prevention, and inform intervention strategies to address disease problems [26,31,35].

Zambia is a low- and middle-income country (LMIC) located in the southern part of the sub-Saharan African region. The first case of COVID-19 in Zambia was reported on 18 March 2020 [45,46]. Since the first report, the identification of SARS-CoV-2 using molecular methods has been reported in subsequent studies [47,48]. A recent study conducted in Lusaka, Zambia, detected adenoviruses and rotaviruses in wastewater [49]. However, there are no published studies that have reported on the detection and identification of SARS-CoV-2 in wastewater systems in Zambia. Given the decline in COVID-19 cases and the paucity of information regarding the current circulation of SARS-CoV-2, this study evaluated wastewater surveillance as a potential tool for monitoring SARS-CoV-2 circulation in Zambia.

## 2. Results

### 2.1. Detection of SARS-CoV-2

A total of nine collections were conducted for the Copperbelt Province and seven for the Eastern Province from October 2023 up to December 2023. A total of 155 samples were processed in the laboratory for SARS-CoV-2 nucleic acid detection. Sample collection per site is summarized in Table 1. Sixty-two (62) samples tested positive for SARS-CoV-2, representing an overall positivity rate of 40% of the study period. Of the positives, the RT PCR Ct value average was 33, while the lowest Ct value was 30.1.

This study found that there was a general trend towards increasing in positivity of wastewater samples for SARS-CoV-2 over the 9-week study time between October and December 2023 (Figure 1).

During the study period, aggregated data from clinical testing for SARS-CoV-2 showed no positive cases in the initial two intervals of wastewater sample collection. Positive clinical cases were recorded only in epidemiological weeks 44, 45, and 46. Clinical cases of SARS-CoV-2 were reported in both the Copperbelt and Eastern provinces during epidemiological weeks 49 and 50. When we compared the wastewater sampling results to clinical cases reported in the respective catchment populations, we noted a potential consistency in an increase in positivity around epidemiological week 44 and week 49, similar to that observed in the wastewater samples (Figure 2).

### 2.2. SARS-CoV-2 Sequencing of Wastewater Samples

From all the 62 SARS-CoV-2 positive samples, sequence data were successfully recovered from 30 samples. The identified reads’ passing filter (PF) was at 89.6% with a Q30 score. Sequencing depth was at 30x achieving 93% and varied across sequences from 144 to 1250x. As expected from wastewater samples, which contain highly fragmented virus genomic material, the sequencing data presented with multiple short fragments of various sequence lengths. However, data from 13 samples with an average genome coverage of 53% were sufficient for confident lineage assignment by Terra and 2 for phylogenetic analysis. The sequences obtained covered a region that corresponds to the spike protein.

All the SARS-CoV-2 variants detected from this study were Omicron subvariants. This study did not detect a specific pattern or trend in variants from the various collection sites. JN.1 was isolated from the Nkana East treatment plant, XBB.1.45 was isolated from the Old Kanini treatment plant, and XBB.2.9 was isolated from the Nkana East treatment plant. BA.2.86 was isolated from the Chipata Motel Ponds and the Kalulushi treatment plant (Figure 3).

The time distribution trends also did not demonstrate any specific pattern or trend. XBB.1.45 was detected in October 2023 from the Old Kanini treatment plant. JN.1 and XBB.2.9 were detected from the Nkana East treatment plant in November 2023. In December 2023, BA.1 and BA.2.86 were detected from wastewater samples collected from the Mindolo Ponds and Kalulushi treatment plants, respectively (Figure 4).

The sequences for the reported clinical cases of SARS-CoV-2 variants collected between October 2023 and December 2023 were downloaded from GISAID. The XBB.1.45 subvariant was detected in wastewater as well as in clinical samples in October 2023. The BA.2.86 subvariant was detected in both wastewater and clinical samples in December 2023. Equally, the BA.5 subvariant was detected in wastewater and clinical samples in November 2023. The JN.1 was isolated in wastewater two weeks before it was reported in clinical cases in the Copperbelt Province. The study also detected other subvariants of public health importance such as BA.1, BA.3, and suspected recombinant variant XM in wastewater, which were absent in clinical cases (Figure 5).

### 2.3. Mutation and Phylogenetic Analysis

The 3495 bp spike gene segment, spanning the region 21651 to 25360 of the SARS-CoV-2 genome, was successfully isolated from a wastewater sample (GISAID accession ID EPI_ISL_19159151) collected at the Mindolo wastewater site. This sequence contained 1.98% ambiguous nucleotides (N) and thirteen mutations: N679K, Y145del, P681H, D614G, H655Y, A570V, N856K, Y144del, E554K, Q954H, P621S, G142D, and V143del. Phylogenetic analysis revealed that the Mindolo/2023 wastewater sequence was distinguishable but clustered within the B.1.1.529 clade (Figure 6). The B.1.1.529 emerged in November 2021 [50]. Meanwhile, the sequence obtained from the wastewater sample in Chipata (GISAID accession ID EPI_ISL_19157057) clustered closely with the BA.5.2.1 lineage, which emerged in early 2022 (Figure 6).

## 3. Discussion

This study was conducted to investigate the presence of SARS-CoV-2 in wastewater in Zambia. To the best of our knowledge, this is the first wastewater surveillance of SARS-CoV-2 being reported in Zambia. Our study found that the overall positivity rate of SARS-CoV-2 was 40% and increased with time from October 2023 to December 2023.

The townships with the highest positivity rates of SARS-CoV-2 in wastewater samples were Nkana East (62%), Mindolo (61%), and New Kanini (46%). This study demonstrated that SARS-CoV-2 was being shed into the wastewater consistently over time. Higher positivity rates of SARS-CoV-2 in wastewater have been reported in previous studies including 50% in Italy [39], 67% in Japan [51], and 81.5% in Pakistan [24]. In comparison to the current study, lower detection of SARS-CoV-2 in wastewater was reported in the USA where the positivity rate was 33.5% [25]. Other lower positivity rates of SARS-CoV-2 detected in wastewater samples have been reported in other studies, including 17% in Ghana [44], 12.28% in Spain [52], and 11.6% in the Czech Republic [38]. Our study findings and those reported in similar studies demonstrate that wastewater surveillance can be used for the early detection of SARS-CoV-2 in communities.

In this study, the detection of SARS-CoV-2 in wastewater is in line with reports concerning clinical cases. For instance, this study detected SARS-CoV-2 RNA in wastewater, which subsequently increased from October 2023 to December 2023. Similarly, when wastewater analysis detected higher levels of SARS-CoV-2, a similar notice was observed in clinical cases reported in the community. As the COVID-19 pandemic rescinded and communities experienced less severe clinical manifestations, their health-seeking behavior to test for COVID-19 as well as clinicians’ demand to request for COVID-19 testing reduced [53]. Notably, our study detected SARS-CoV-2 from the first collection, suggesting a sustained viral shedding pattern from the communities despite the low to no clinical cases of COVID-19 being reported at that time. This is similar to the findings from Denmark where SARS-CoV-2 variants were detected from wastewater samples despite the discontinuation of testing for COVID-19 at the community level [54]. During our study, an appreciable increase in the detection of SARS-CoV-2 in wastewater was noted and increased with time. Some studies have demonstrated that upon identification of SARS-CoV-2 in wastewater samples, the detection increased with time similar to what has been reported in clinical cases and suggests that wastewater surveillance can be used as an early warning to predict positive clinical cases for SARS-CoV-2 in communities before clinical presentations and detection [39,51,55,56]. A Japanese study found a lag period of 3 and 7 days, indicating the importance of wastewater surveillance in the early detection and spread of SARS-CoV-2 [51]. This indeed highlights the potential of wastewater surveillance as an early indicator of COVID-19 outbreaks and emphasizes the importance of incorporating this method into existing disease surveillance systems [29,41,55,57].

Our study detected Omicron as the main variant in wastewater across the sampled communities. The Omicron variant was first discovered in South Africa [58]. Its spread and clinical features affecting many countries in Southern Africa have been reported [59]. Consequently, many variants of Omicron have surfaced [60]. Previous studies that were conducted in Zambia revealed the presence of various variants of SARS-CoV-2 among clinical cases including Alpha, Beta, Delta, and many Omicron variants among clinical cases [47,48]. In Zambia, a previous study demonstrated that most variants were few, with only Omicron being the majority [47]. A recent study conducted in Zambia only detected Omicron in clinical cases, potentially indicating the replacement of other variants [61]. It does seem that Omicron dominated and replaced other variants and our study findings corroborate with those reported from studies that found the Omicron variant in the wastewater samples [39,51]. An Egyptian study found that the Delta variant was replaced by the Omicron variant, which became a predominant variant [62], similar to findings from Puerto Rico [63] and the USA [64]. Similarly, a study in Italy found that wastewater samples tested positive for the SARS-CoV-2 Omicron variant on 7 December 2021, and increased to 66% in the last week of December 2021, indicating the predominance of Omicron and spread across the country [39]. Yet another study in Mexico also reported that Omicron became a predominant variant and replaced the Delta variant [65]. Understanding the predominant variants is very critical in guiding mitigation strategies to address COVID-19 [66,67].

The present study detected the BA.5 subvariant in wastewater samples, which was also identified in clinical cases. Again, our findings are consistent with other studies that reported the detection of BA.5 in wastewater in Germany [68] and Italy [69,70]. Other studies, however, have reported contrasting results showing the presence of subvariants other than BA.5 during the same period [71]. The differences could indicate a possible temporal shift in the predominance of variants with replacements of the BA.5 by other variants with time. Our study also detected XBB variants in wastewater, with the first detection of XBB.1.4.5 on 19 October 2023, which was also reported in clinical cases. The detection of these variants in wastewater indicates that there was indeed circulation of these variants. This is similar to findings from other countries where they detected XBB variants circulating in wastewater [72]. Another study in Italy detected XBB subvariants in November 2022 [71].

In Zambia, the BA.2.86 subvariant was detected in wastewater samples on 7 December 2023, in this study. The BA.2.86 was also identified in clinical cases in Zambia. The first global report of the BA.2.86 variant was in August 2023 [73,74]. The presence of BA.2.86 in wastewater samples is due to the increased shedding of the virus in fecal material [75]. Our findings corroborate with other studies that also detected SARS-CoV-2 BA.2.86 in wastewater and clinical cases, including reports from Sweden [76], Thailand [77], China [78], and Israel [79].

The highly mutating BA2.86 evolved into JN.1 and other subvariants [80]. The JN.1 subvariant was first reported in Luxemburg/ Iceland on 25th August 2023 and was initially tracked as part of BA2.86 [81]. However, its continued report in many countries led to it being the majority among the BA2.86-descendent lineages. The growth advantage of the JN.1 subvariant and its ability for antibody escape, increasing its potential for lethal clinical outcomes, led to the WHO classifying it as a variant of interest separate from BA2.86 [25,82]. The first clinical report of the JN.1 variant in Zambia was on 4th December 2023 in Nakonde, Muchinga Province. The present study reported the presence of the JN.1 variant in wastewater from the Copperbelt Province earlier on 24 November 2023, which was followed by its detection in clinical cases on 12 December 2023 in the same province. This coincided with reports of spikes in COVID-19 cases in Zambia from December 2023 to January 2024. Similar findings were reported in India and Germany where the JN.1 became the predominant variant towards December 2023, replacing the BA.2.86 and other subvariants [83,84].

The presence of possible recombinant strains (XM and GE) in this study allows us to speculate that there could be active SARS-CoV-2 recombination among circulating variants. Therefore, the continued circulation of the BA.1 variant may be dormant in the environment and yet allows us to assert its possible contribution to ongoing recombination, posing a risk of brewing more virulent strains. Furthermore, this ‘dormant’ variant could be a source for future mutated versions of clinical significance even without recombination. Hence, it is key to strengthen environmental surveillance to detect the dormant variant in the environment as part of viral mutation vigilance for outbreak preparedness.

Our study found no unique mutations observed in the Mindolo wastewater sequence. However, it exhibited an evolutionary mutation signature: A570V, D614G, G142D, H655Y, P681H, V143del, Y144del, and Y145del, which can be classified as rare, indicating that it only occurred once. Interestingly, phylogenetic analysis positioned this sequence from the Copperbelt Province in the B.1.1.529 clade, suggesting that earlier Omicron variants detected in late 2021 could still be circulating and may not have been wholly replaced by newer subvariants. It seems reasonable to assert that the B.1.1.529 may not have been completely replaced by newer Omicron subvariants and thus could be circulating and possibly contributing to the community transmission of SARS-CoV-2. It is unlikely that the SARS-CoV-2 RNA integrity may have been maintained for close to three years in wastewater. It is important to note that this sequence may not fully represent the viral composition at the Mindolo wastewater site, as several other lineages were identified within the same sample. As for the sequence obtained from the Chipata Motel Pond, it could not be assigned to a specific lineage because it contained a high proportion of ambiguous nucleotides in its sequence. However, phylogenetic analysis placed it close to the BA.5 series of Omicron subvariants.

We acknowledge the limitation of our study, particularly the small sample size, as it only covered two provinces across Zambia and the short duration of the study, thereby rendering a weakness to detect time-seasonal trends. However, our findings are of public health importance as they demonstrate the utility of monitoring SARS-CoV-2 in wastewater in LMICs, including Zambia.

## 4. Materials and Methods

### 4.1. Study Design and Sites

This was a prospective longitudinal study that collected samples between October 2023 and December 2023 from the Copperbelt and Eastern provinces in Zambia. The study sites included Ndola district (Old Kanini, New Kanini, and Lubuto), Kitwe (Nkana East, Mindolo, Chambishi, and Kalulushi) (Figure 7A), and Chipata (Chipata Motel and Chipata ponds) (Figure 7B). A total of nine weekly collections were conducted for Ndola and Kitwe while seven collections were conducted for Chipata. The sampling was conducted at influent points of sewer ponds and treatment plants that service selected parts of the districts.

### 4.2. Sample Collection

A composite–grab sample of one liter was collected from each site, with the aid of a 2-litre bucket, as described previously [25]. A bag-mediated filtration system was also used to collect samples from each township. Sample collection was conducted following strict safety procedures safeguarding the sample and study staff. Collected samples were immediately stored at 4 to 8 °C using cool boxes with controlled and monitored temperatures and mobile vehicle refrigerators during transportation. Samples were processed immediately as soon as they were received at the Churches Health Association of Zambia (CHAZ) laboratory in Lusaka.

### 4.3. Viral Concentration

The wastewater samples were subjected to three (3) viral concentration methods before nucleic acid extraction. These included the skimmed milk flocculation, bag-mediated filtration system, and polythene glycol-based (PEG) concentration method.

#### 4.3.1. Skimmed Milk Flocculation

A composite wastewater sample (500 mL) was used for the direct skimmed milk flocculation method. Briefly, 0.5 g of skimmed milk powder was dissolved in 50 mL of sterile water to obtain 1% (*w*/*v*) skimmed milk solution [85]. The pH of the solution was carefully adjusted to 3.5 using 1M HCl. We added 5 mL of a 1% skimmed milk solution to 500 mL of the wastewater composite sample to attain a final concentration of 0.01% (*w*/*v*). Samples were stirred for 8 h, and flocs were allowed to settle at room temperature for another 8 h. The supernatant was carefully removed using serological pipettes without agitating the flocs. A final volume of 50 mL containing the flocs was transferred to 50 mL falcon tubes and centrifuged at 8000× *g* for 30 min. The supernatant was carefully removed, and the pellets were carefully dislodged. The pellets were resuspended in 2 mL of 0.2 M phosphate buffer saline (PBS), pH 7.5. The final concentrated sample was aliquoted in small working volumes (~200 μL) and stored at −80 °C until viral nucleic acid extraction.

#### 4.3.2. Bag-Mediated Filtration System (BMFS)

Samples were collected into the BMFS bag and processed, as described previously [86]. Briefly, wastewater (6000 mL) from the composite sample was collected in sampling bags with a pre-screen mesh (249 μm pore size) over the opening. The composite sample was filtered into the collection bag mounted on a custom-made tripod stand (BoundaryTEC, Minneapolis, MN, USA). A ViroCap^TM^ filter (Scientific Methods, Inc., Granger, IN, USA) was attached to the bag’s outlet port and the sample was allowed to flow through the filter by gravity. The average volume filtered was about 5400 mL, over an average of 90 min to 3 h (the time for filtration was dependent on the amount and size of fecal debris in the wastewater). The filter was transported under a cold chain to the CHAZ laboratory for viral elution and secondary concentration with skimmed milk. To elute the virus from the viral cup filter, the beef extract eluate was injected into the filter inlet and incubated for 30 min before recovering the eluate through the filter outlet using a peristaltic pump.

##### Secondary Concentration Procedure

A total of three (3) mL of 1% *w*/*v* skimmed milk was added to the eluate. The pH of the mixture was adjusted to the range of 3.0–4.0 using sodium hydroxide (NaOH) and hydrochloric acid (HCl). To coagulate the proteins in flocs, the eluate was incubated at room temperature and shaken evenly for 2 h. The sample was aliquoted into 50 mL conical tubes and centrifuged at 3500× *g*, for 30 min at a temperature of 4 °C, and thereafter returned to the biosafety cabinet (BSC) where the supernatant was decanted. The pellets were completely resuspended in 10 mL of sterile PBS, pH 7.4 by vigorous vortexing. The concentrated sample was aliquoted in 1 mL working volumes and stored at −80 °C until use.

#### 4.3.3. Polyethylene Glycol-Based (PEG) Concentration

The PEG precipitation was performed as previously described [87]. Briefly, 14 g of PEG 8000 and 1.17g of NaCl were added to 100 mL of the composite sample. The PEG 8000 and NaCl were completely dissolved by vigorous shaking. The sample was stirred at 200 rpm, for 4 h at 4 °C followed by centrifugation at 6500× *g*, for 30 min at 4 °C. The supernatant was discarded and the pellet was resuspended in 6 mL of sterile PBS (pH 7.4). The final concentrated sample was aliquoted in small working volumes (~200 μL) and stored at −80 °C until use.

### 4.4. Nucleic Acid Extraction 

Viral RNA was extracted from the 155 wastewater concentrated samples using the MagMAX viral isolation kit (Applied Biosystems, Foster City, CA, USA) on an automated Kingfisher Flex 96 Deep-well magnetic particle processor (Thermo-Fisher Scientific, Carlsbad, CA, USA) according to the manufacturer’s recommendation. Briefly, in a class II biosafety cabinet, 200 µL of wastewater concentrated sample was added to 265 µL of lysis/binding solution. An amount of 20 µL of bead mix was then added to the sample–lysis buffer mix. Following the removal of the supernatant, samples were washed twice with wash buffer and RNA was eluted in 50 µL of elution buffer and stored at −80 °C.

### 4.5. SARS-CoV-2 Detection by RT-PCR

To confirm the presence of SARS-CoV-2 nucleic acid in wastewater RNA samples, the TaqPath COVID-19 CE-IVD RT-qPCR master mix (Thermo-Fisher Scientific, Waltham, MA, USA) was used. The 25 µL reaction mix contained 15 µL of TaqPath qPCR master mix and 10 µL of extracted RNA. The TaqPath COVID-19 CE-IVD RT-qPCR kit mix has negative, positive, and internal controls to monitor the reliability of the results from RNA extraction to PCR amplification. As per protocol, the internal control with a Ct ≤ 37 was considered positive for SARS-CoV-2. Additionally, N, ORF1ab, and S genes with Ct ≤ 37 were considered positive for SARS-CoV-2. In this study, the Ct cutoff of ≤37 was considered positive while samples with no amplification (Ct = 0) for any of the three targets (N, ORF1ab) excluding the S gene (due to mispriming or primer hybridization failure due to mutations) were considered as not valid. The reporter dye and detector dye were set on the ABI 7500 Fast Real-Time PCR system (ThermoFisher, Carlsbad, CA, USA) as follows: FAM: ORF1ab, VIC: N gene, ABY: S gene, JUN: MS2. A positive and no-template control was added to each run. The samples were incubated for 2 min at 25 °C, 10 min at 53 °C, and 2 min at 95 °C followed by amplification of 40 cycles consisting of 3 s at 95 °C and 30 s at 60 °C. Fluorescence detection was set during the annealing/extension step (30 s at 60 °C). All samples that tested positive for SARS-CoV-2 positive samples were considered for whole genome sequencing.

### 4.6. Genomic Sequencing

#### 4.6.1. cDNA Synthesis and Amplification of SARS-CoV-2

Random hexamer priming using the first-strand cDNA master mix (Illumina) achieved first-strand cDNA synthesis according to the manufacturer’s recommended protocol. In a 96-well PCR plate, 8.5 µL of random hexamers was added to 8.5 µL of extracted RNA and denatured on an ABI 7500 real-time PCR for 3 min at 65 °C and then incubated at 4 °C. Ten microliters of first-strand mix and one µL of reverse transcriptase were then added to the denatured sample. cDNA synthesis was achieved with the following conditions: 5 min at 25 °C, 10 min at 50 °C, and 5 min at 80 °C. SARS-CoV-2 genome amplification was conducted using ARCTIC network V4 primer pools (https://github.com/artic-network/primer-schemes, accessed on 1 January 2024; provided by Illumina). ARCTIC V4 primer pool amplification employed two reactions per sample, i.e., COVIDseq Primer Pool 1 (CPP1) and COVIDseq Primer Pool 2 (CPP2). The reaction components for each reaction consisted of 12.5 µL Illumina PCR Master Mix, 3.5 µL of either CPP1 or CPP, 5 µL of first-strand cDNA synthesis, and 3.9 µL of nuclease-free water. Thermoprofiles were as follows: 1 cycle of 98 °C for 3 min, followed by 35 cycles of 98 °C for 15 s and 63 °C for 5 min. For each run, a single positive control (TaqPath COVID-19 Control; ThermoFisher, Carlsbad, CA, USA) and negative no-template control (nuclease-free water) were included to serve as indicators of extraneous nucleic acid contamination. PCR amplicons for each sample were then combined by transferring 10 µL from each well of the CPP1 and CPP2 into a new well.

#### 4.6.2. Library Preparation and Illumina Sequencing

Library preparation was performed using the Illumina COVIDseq kit (Illumina Inc., San Diego, CA, USA) on the automated liquid robotic handler instrument (Hamilton, NV, USA). Pooled PCR products were processed for tagmentation and adapter ligation using the Illumina COVIDseq Kit with IDT Illumina-PCR indexes. Pooling and library clean-up were performed as per the protocol provided by the manufacturer (Illumina Inc.). Pooled libraries were quantified using the Qubit 4.0 fluorometer (Invitrogen Inc., Waltham, MA, USA) using the Qubit dsDNA High Sensitivity kit (Life Technologies, Eugene, OR, USA). The pooled library was normalized to a 4 nM concentration. The normalized library (4 nM) was denatured and neutralized using 0.2 N sodium hydroxide and 200 mM Tris-HCL (pH7), respectively. The library was further diluted to a final loading concentration of 0.5 nM and sequenced (301 paired-end) on the Illumina NextSeq 2000 (Illumina, SanDiego, CA, USA) platform.

#### 4.6.3. Genome Assembly and Annotation

To assemble SARS-CoV-2 whole genomes, the Illumina DRAGEN DNA pipeline was used to analyze the sequence reads prepared. The DRAGEN pipeline uses a kmer reference database to match kmers from the sequencing read to kmers from the SARS-CoV-2 reference genome (Wuhan Hu-1, accession no. NC_045512). The DRAGEN BCL converter was used to generate the FASTQ files, which were analyzed using a SARS-CoV-2 specific wastewater sequence data analysis tool called Freyja, available on a cloud-native platform, Terra Bio (Version 1.3.0 [Freyja] Retrieved 14 May 2024 from https://app.terra.bio/#workspaces/aphl-personal/aphl-zambia_wwbs_workstation3/ Freyja). Freyja measured the sequenced genome to return an estimate of the true lineage abundances in each sample. This method used lineage-defining “barcodes” derived from the UShER global phylogenetic tree as a base set for demixing. After this, it further called the variants and captured sequencing depth information to determine the relative abundance of lineages present.

### 4.7. Data Analysis

Demographic data were entered into Microsoft Excel version 2013, cleaned then exported to Stata version 14 (Stata Corp, College Station, TX, USA) for statistical analysis. The prevalence of SARS-CoV-2 in wastewater was presented as proportions and their 95% confidence intervals. Data were summarized and visualized using the R package dplyr v1.0.7 (Rstudio, Boston, MA, USA).

### 4.8. Phylogenetic Analysis

Mutation characterization involved utilizing the CoVsurver tool accessed through GISAID (https://gisaid.org/database-features/covsurver-mutations-app/, accessed on 23 May 2024). 

For phylogenetic analysis, AudacityInstant, accessed through GISAID, was initially used to retrieve SARS-CoV-2 genome sequences from GISAID that closely resembled the sequences generated in this study. These sequences, along with the sequences obtained in this study, underwent multiple sequence alignment using Clustal Omega [88]. Subsequently, the aligned sequences were trimmed to a length of 3495 bp using a custom Python script (version 3.6). A maximum likelihood phylogenetic tree was then constructed using the Kimura-3-parameter model with unequal base frequencies, estimation of base frequencies, and estimation of the proportion of invariable sites (K3Pu+F+I), as determined by the model test integrated into IQ-TREE (http://www.iqtree.org/, accessed on 23 May 2024). The phylogenetic tree construction in IQ-TREE included 1000 bootstrap replicates to ascertain genetic tree reliability. The resulting maximum likelihood tree was rooted using TempEst v1.5.3 (http://tree.bio.ed.ac.uk/software/tempest/, accessed on 23 May 2024), which estimated the best-fitting root of the phylogeny by minimizing the variance of root-to-tip distances using the heuristic residual mean squared function. Finally, the ML tree file underwent editing using the Interactive Tree of Life (iTOL) v5 (https://itol.embl.de/, accessed on 23 May 2024), an online tool designed for the display and annotation of phylogenetic trees.

### 4.9. Collection of Clinical Data for COVID-19

Country data for COVID-19 clinical cases were reviewed from Worldometer (Zambia COVID-Coronavirus Statistics–Worldometer, https://www.worldometers.info/ accessed on 17 May 2024) as well as from the districts covering the study sites for the period October 2023 to December 2023. Sequencing data for clinical samples during the study period were obtained from GISAID (GISAID Initiative https://gisaid.org/ accessed on 17 May 2024), where 17 sequences were downloaded.

## 5. Conclusions

This study demonstrated the presence of SARS-CoV-2 in the wastewater samples from several districts in Zambia. The detection of the JN.1 Omicron variant was a critical public health finding, as it was a globally recognized variant of interest, indicating its circulation in some surveyed communities in Zambia. There is a need to strengthen wastewater surveillance in the monitoring of SARS-CoV-2 in communities in Zambia. Laboratory-based molecular detection of SARS-CoV-2 in the collected wastewater samples in this study indicated a successful ‘proof of concept’ for our wastewater surveillance activities in Zambia. Therefore, strengthening wastewater surveillance of SARS-CoV-2 is critical in identifying circulating SARS-CoV-2 variants and other threats that may emerge in the future.

## Figures and Tables

**Figure 1 ijms-25-08839-f001:**
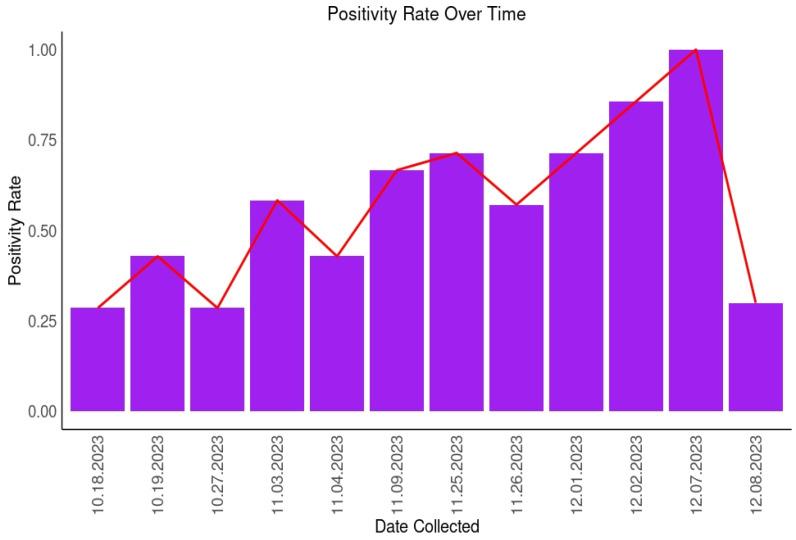
Positivity trends of SARS-CoV-2 in wastewater over the data collection period (October 2023 to December 2023).

**Figure 2 ijms-25-08839-f002:**
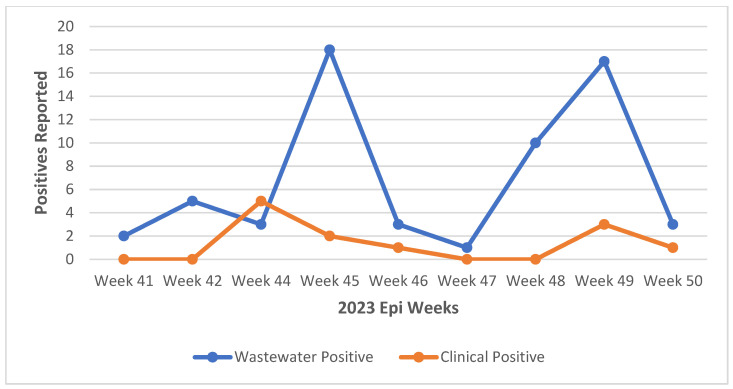
SARS-CoV-2 positivity trends in wastewater and clinical cases.

**Figure 3 ijms-25-08839-f003:**
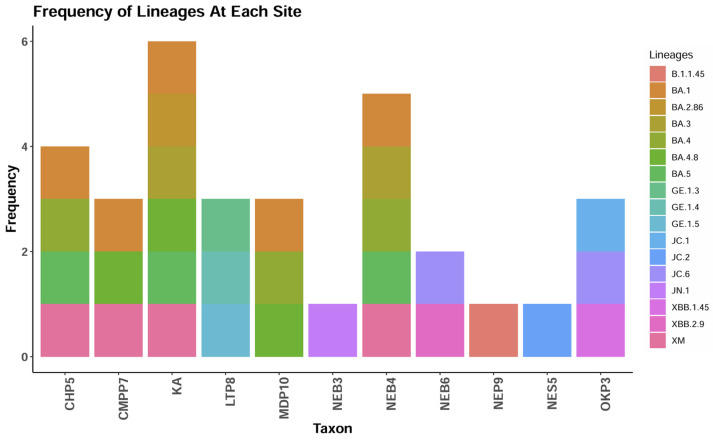
SARS-CoV-2 variants detected from wastewater samples per study sites. The key for sampling sites abbreviations is as follows: CHP5—Chipata Pump Station; CMPP7—Chipata Motel Ponds; KA—Kalulushi; LTP8—Lubuto; MDP10—Mindolo; NEB3,4,6,9—Nkana East; NES5—Nkana East treatment plant; OKP3—Old Kanini.

**Figure 4 ijms-25-08839-f004:**
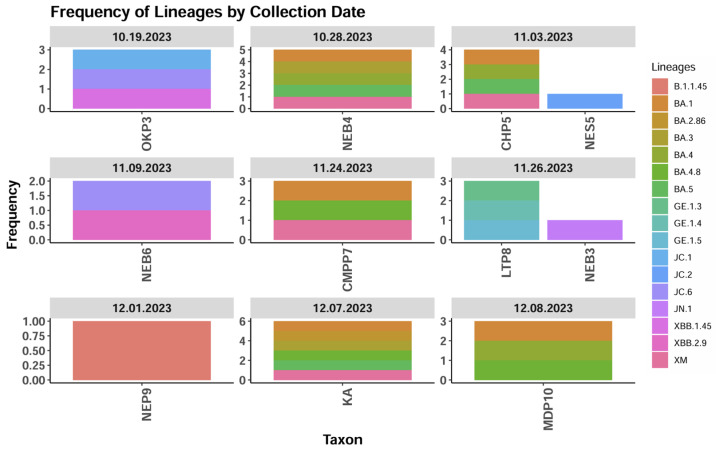
SARS-CoV-2 variants isolated per sampling period. The key for sampling sites abbreviations is as follows: CHP5—Chipata Pump Station; CMPP7—Chipata Motel Ponds; KA—Kalulushi treatment plant; LTP8—Lubuto treatment plant; MDP10—Mindolo Ponds; NEB3,4,6—Nkana East treatment plant; NEP9—Nkana East treatment plant; NES5—Nkana East treatment plant; OKP3—Old Kanini treatment plant.

**Figure 5 ijms-25-08839-f005:**
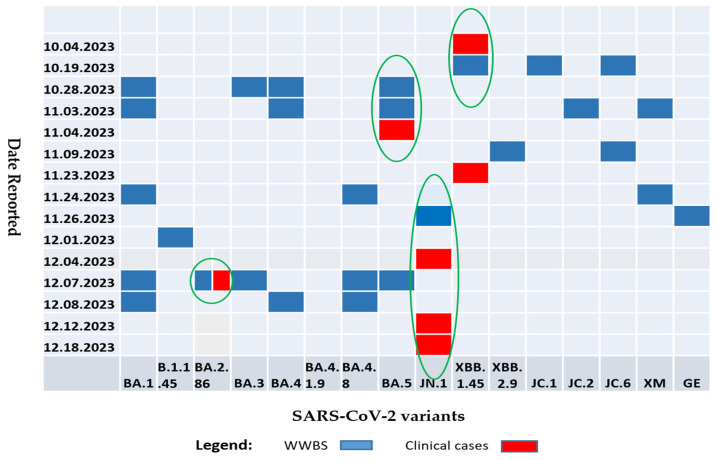
Sequencing of SARS-CoV-2 results comparing wastewater-based surveillance (WWBS) variants (blue shade) versus reported Omicron subvariants in clinical cases.

**Figure 6 ijms-25-08839-f006:**
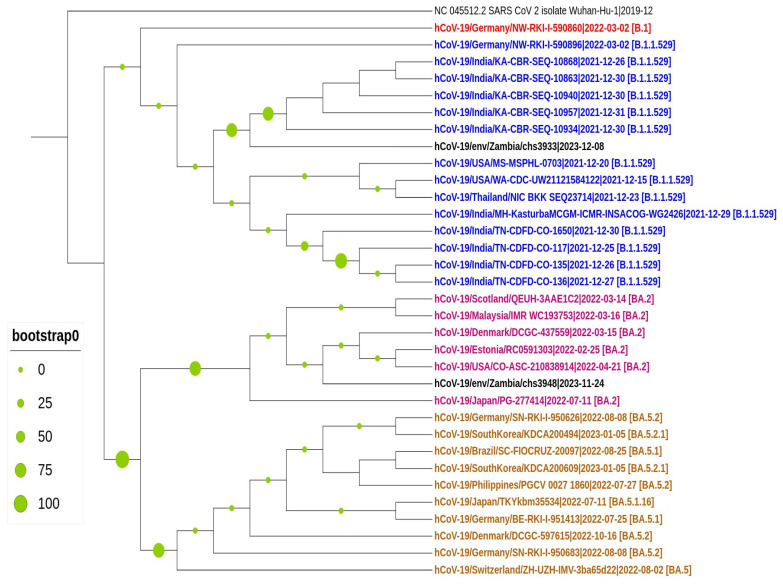
Maximum likelihood phylogenetic analysis of SARS-CoV-2. Sequences generated in this study are highlighted in black. The tree scale indicates the number of nucleotide substitutions per site.

**Figure 7 ijms-25-08839-f007:**
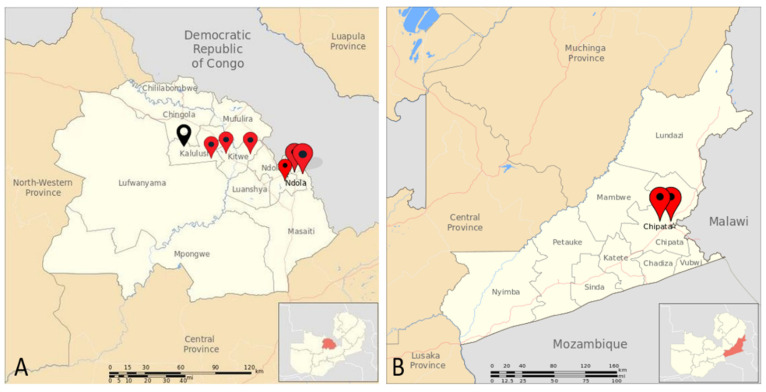
Geographical map showing sampling sites for (**A**) Copperbelt and (**B**) Eastern provinces of Zambia. The red pins with black circles represent eight sampling sites with nine and seven collections while the black pin with a light-yellow circle represent one site with one collection.

**Table 1 ijms-25-08839-t001:** The positivity rate of wastewater samples for SARS-CoV-2 per study site.

Site (n)	Positive n (%)	95% CI
Chambishi (n = 18)	7 (39%)	17.3–64.3
Chipata Motel Ponds (13)	2 (15%)	1.9–45.4
Chipata Motel Pumps (20)	3 (15%)	3.2–37.9
Lubuto (18)	6 (33%)	13.3–59.0
Mindolo (18)	11 (61%)	35.7–82.7
New Kanini (26)	12 (46%)	26.6–66.6
Nkana East (26)	16 (62%)	40.6–79.8
Old Kanini (16)	5 (31%)	11.0–58.7
Total (155)	62 (40%)	32.2–48.2

## Data Availability

The data presented in the study are available on request from the corresponding author.

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
