# Peer review of "Wastewater Surveillance of SARS-CoV-2 in Zambia: An Early Warning Tool"

_ijms, 2024, doi:10.3390/ijms25168839_

Round 1

Reviewer 1 Report

Comments and Suggestions for Authors

1. The study was conducted in only two provinces, limiting the sample coverage and making it challenging to generalize the findings to the entire country.

2. The short study duration does not fully capture the effectiveness and advantages of long-term wastewater monitoring for pandemic surveillance.

3. Some figures lack clarity and detailed annotations, affecting the visual appeal and readability of the results. 

4. Several paper relate to wastewater suggested to be read. (e.g. ECOL CHEM ENG S. 2014, 21(1):89-99; Sustainability 2017, 9, 1845)

Comments on the Quality of English Language

Moderate editing of English language required. 

Reviewer 2 Report

Comments and Suggestions for Authors

Shempela et al. report their findings after a longitudinal study applying viral enrichment and sequencing methods to wastewater samples collected from two regions in Zambia.  Their analysis focuses on detection of SARS-CoV2 variants, by performing sequencing only of samples that are positive for SARS-CoV2 by RT-qPCR, using a kit that theoretically targets the sequences of well-conserved regions of the coronavirus spike (S), nucleocapsid (N), and Orf1ab (ORF1ab) proteins. Downstream sequence analyses were simplified by only performing analysis of reads that map to a reference coronavirus using Illumina’s DRAGEN DNA workflow.

As best I can tell, the novelty of this paper relies on it being the first longitudinal SARS-CoV2 detection study to be published out of Zambia. The methods used for viral enrichment (milk flocculation or PEG precipitation) are standard protocols. Use of RT-qPCR to focus sequencing on the samples most likely to contain the target(s) of interest (e.g., SARS-CoV2 particles) is a clever way to reduce sequencing costs, but also not new. To be clear, there’s nothing inherently wrong with this – it is of great public health importance to understand how locally circulating viral variants compare with the dominant regional or global strains. But I have some concerns with the analyses performed and some of the conclusions drawn from them.

The first concern is that the authors rely on DRAGEN to identify coronavirus reads by comparison with only a single reference genome. I’m not suggesting that they should use all reference coronavirus sequences, but I think most human lineages can be well-represented by a dozen or two reference isolates, and using a more inclusive set of references would likely enhance the sensitivity of the authors’ workflow.

The second concern is that the authors have not provided enough information about the read processing or read/contig statistics (i.e., quality filtration, read correction, assembly details, number of SARS-CoV2 contigs assembled per sample, contig length range per sample, read depth across those contigs, etc.) for me to know how much I trust their lineage sorting based on sub-genomic fragments. Was sequencing done at sufficiently high depth, and assembly performed in a manner that would allow one to know whether multiple coronavirus variants were present in a single sample? Or is this Illumina-based amplification and sequencing workflow really intended to converge at a single consensus sequence regardless of the sample composition? The authors state that they have 13/30 (43%) sequenced samples with an average genome coverage of 53% spanning the spike protein region (which contains most of the mutations used for lineage sorting), but for accurate lineage sorting it matters whether that coverage comes from a single contig or multiple contigs. Especially because the overlapping tiled sequencing approach used here should amplify all sub-genomic regions from all variants present in a sample. Another question – from what data did the authors conclude they have a possible recombinant variant (XM)? I am not aware of this lineage – is it recently discovered? Is it novel to this study? What data support its existence here? This is never really explained in the manuscript.

Finally, the presentation of identified lineages/mutants is very surface level. If 13 samples have enough genome coverage for lineage sorting, why are only two samples shown in Figure 7? If the average genome coverage of 53% spans the spike protein for the 13 samples, why was only one spike gene segment analyzed for mutations, and which reference was chosen for that analysis, and why? Overall, a lot of missing pieces for me – these are just the most egregious examples.

Round 2

Reviewer 1 Report

Comments and Suggestions for Authors

It can be accepted. 

Author Response

Dear Reviewer. Thank you so much for the recommendation to accept our paper.